# Silica-Fiber-Reinforced Composites for Microelectronic Applications: Effects of Curing Routes

**DOI:** 10.3390/ma16051790

**Published:** 2023-02-22

**Authors:** Imran Haider, Iftikhar Hussain Gul, Malik Adeel Umer, Mutawara Mahmood Baig

**Affiliations:** Thermal Transport Laboratory, Department of Materials Engineering, School of Chemical and Materials Engineering (SCME), National University of Sciences and Technology (NUST), Islamabad 44000, Pakistan

**Keywords:** reinforced composites, microwave curing, dielectric properties, thermal properties

## Abstract

For curing of fiber-reinforced epoxy composites, an alternative to thermal heating is the use of microwave energy, which cures quickly and consumes less energy. Employing thermal curing (TC) and microwave (MC) curing methods, we present a comparative study on the functional characteristics of fiber-reinforced composite for microelectronics. The composite prepregs, prepared from commercial silica fiber fabric/epoxy resin, were separately cured via thermal and microwave energy under curing conditions (temperature/time). The dielectric, structural, morphological, thermal, and mechanical properties of composite materials were investigated. Microwave cured composite showed a 1% lower dielectric constant, 21.5% lower dielectric loss factor, and 2.6% lower weight loss, than thermally cured one. Furthermore, the dynamic mechanical analysis (DMA) revealed a 20% increase in the storage and loss modulus along with a 15.5% increase in the glass transition temperature (Tg) of microwave-cured compared to thermally cured composite. The fourier transformation infrared spectroscopy (FTIR) showed similar spectra of both the composites; however, the microwave-cured composite exhibited higher tensile (15.4%), and compression strength (4.3%) than the thermally cured composite. These results illustrate that microwave-cured silica-fiber-reinforced composite exhibit superior electrical performance, thermal stability, and mechanical properties compared to thermally cured silica fiber/epoxy composite in a shorter time and the expense of less energy.

## 1. Introduction

Fiber-reinforced composites (FRC) are superior to other structural materials, due to their high specific strength and stiffness, high temperature, and fatigue resistance [1,2]. Glass-fiber-reinforced composites have outstanding properties, easy manufacturing, and reasonable cost compared to other fiber-reinforced composites [3,4]. The mechanical properties depend on the fiber as well as the matrix properties. The physical and mechanical properties of the thermoset resins, such as epoxy, depend on curing conditions [5]. The mechanical performance of FRC is influenced by the properties of the phases and their interactions. The interface and fiber-matrix bonding are quite important for enhancing the effective mechanical properties of FRC [6,7]. Amorphous materials (polymers, glasses, and metals) change their structural state below Tg, as physical aging or structural relaxation affect the properties of the material (mechanical, dielectric, and thermal behavior) [8]. The epoxy resin in a B-level semi-cured state can be cured under specific conditions to obtain high-quality composite products [9]. The curing degree of the resin has a very large effect on the properties of the composite [10,11,12].

In adhesively bonded structures, the quality of bonded composites has a strong relationship with the variabilities caused by process parameters, such as temperature, curing duration, and rate [13,14]. Usually, ambient, thermal curing, or a combination of both is used, depending on the requirement. As the demand for lighter, cheaper, and more compact electronic devices increases, there is a greater need to develop innovative and fast processing techniques with higher energy efficiency and reduced cure time. Longer curing times in thermal curing reduce the production throughput. In the electronics industry especially, the curing of thermoset systems has become a limiting factor in the production time; other processing methods are needed to be explored to reduce manufacturing costs or increase energy efficiency. One such method makes use of microwave radiation to heat epoxy resins [15]. The microwave curing process (MCP) can reduce production costs [16]. The MCP mechanism operates through dipolar loss, conduction loss, hysteresis loss, and Eddy current loss [17]. For thermosetting polymers, microwave curing involves the microwave radiation to heat the sample directly, which leads to efficient, fast, and selective heating, compared to conventional thermal curing, where the samples are heated indirectly. Microwave radiation also heats in a volumetric manner, thus leading to less temperature disparity and therefore a less severe temperature gradient within the material, leading potentially to less internal residual stress [18,19,20]. Microwave curing produces composites with mechanical properties comparable to those produced with the autoclave process while reducing 45% processing time and 3% energy consumption [21]. The curing kinetics has a significant influence on the macroscopic mechanical properties of composite materials [22,23,24]. Comparative studies of microwave-cured and thermally cured composites have reported contradictory results, while many authors have shown a reaction rate enhancement for microwave curing compared to thermal curing [25,26].

For many applications, the advantages of microwave curing must not be outweighed by a loss of thermal, chemical, or mechanical properties. Microwave heating is known as the most efficient volume-heating process due to its excellent depth of penetration in polymers [27]. Its process can be tuned with the use of highly efficient dielectric nanomaterials in polymers to boost micro-level heating at the molecular scale [28,29,30]. Dielectric nanomaterials can efficiently absorb radiation and convert it to molecular vibrations/rotations via dipole moments, mainly because of dipolar polarization. The vibrations can then add to the heating level in the polymer surrounding the nanomaterials by the friction mechanism [29].

The mechanical properties of certain microwave-cured materials are similar or even increased compared to conventional cures [27,28,29,30,31]. Generally, an increase in the glass transition temperature, Tg, has been reported [32]. However, depending on the type of curing agent used, Tg can be seen to decrease compared to thermally cured samples [33]. When composites are used as interconnections, printed circuit boards, and airplane skin materials, their dielectric properties become very important and must be determined before they can be used in these applications [34]. For polymer composites, the dielectric properties are associated with the component fractions [35,36] and they are widely used in electronics applications because of their good dielectric and mechanical properties. Silica fibers are used as reinforcement due to their small coefficient of thermal expansion, low thermal conductivity, superior mechanical strength, and excellent dielectric properties [37]. Compression molding, hand lay-up, spray-up, vacuum infusion, vacuum bagging, resin transfer molding (RTM), autoclave molding, filament winding, automated fiber placement (AFP), pultrusion, injection molding, vacuum forming, and stamp forming are most of the manufacturing techniques used in composite production. Prepegging produces a semi-finished composite by controlling the curing to fulfill the final requirement [38]. The curing process parameters have a crucial influence on the quality of the composite products [39]. Effects of curing degree, of Quartz fiber-boron phenolic composites, were investigated by the mechanical properties test, scanning electron microscope (SEM), and thermogravimetric analysis [40]. Fourier-transform infrared analysis shows no significant difference between the conventional and microwave cured samples [41]. The chemorheology of a filled epoxy system declares isothermal DSC measurements to be inadequate in the case of fast-curing thermosets [42]. Generally, a comparison between literature data is difficult due to the variety of curing agents used and their effects on the curing of the epoxy resin [43]. As preferred by many researchers, one of the most common methods is the mechanical testing of composite properties on different cure regimes, which can optimize the curing parameters [44].

Different composite curing studies have investigated various aspects of functional fiber-reinforced composites, but in microelectronics, low dielectric constant and loss factor of material is a matter of interest. In this study, multiple proportions of silica fiber/epoxy composite (prepregs) were cured thermally, via microwave energy, and their dielectric, structural, morphological, thermal, and mechanical properties were investigated.

## 2. Materials and Methods

### 2.1. Materials

Commercial silica fiber fabric BWT260-82 (*ρ* = 2.25 g/cc, avg. fiber dia = 6.86 μm, UTS = 1.8 GPa, SiO_2_ ~95%, thickness = 0.21 mm) was obtained from Business and Engineering Trends, Punjab, Pakistan (BET Pakistan). The bonding adhesive used was a two-component commercial Epoxy RER160 and curing agent REH160 (RESSICHEM, Karachi, Pakistan). Commercial ethanol was used as a solvent in the resin matrix preparation.

### 2.2. Preparation of Composite Prepregs

Unidirectional woven silica fabric was cut into fabric plies (200 × 200 mm), washed with commercial ethanol, air-dried (30 min), and then oven-dried (30 min) at 120 °C. Lab environmental conditions were 25 °C and 36% RH. In a 1000 mL glass beaker, epoxy (500 gm) and hardener (50 gm) were weighed (SHIMADZU-UW 3400 g, Kyoto, Japan), and the resin matrix was prepared with dropwise addition ethanol (30 mL). The resin matrix was poured on fabric plies and distributed with an applicator. By hand lay-up, the impregnated plies were stacked layer by layer between clean surfaces of mold plates and retained for 2 h. Stacked laminates (2 No.) were simultaneously processed to form composite prepregs (S0.3E0.7_,_ S_0.4_E_0.6_, S_0.5_E_0.5_, S_0.6_E_0.4_, S_0.7_E_0.3_) where “S” refers to silica fabric, “E” refers to epoxy resin and the subscript refers to the wt. fraction. The fabrication process (raw material to prepreg to composite) is shown in Figure 1.

### 2.3. Curing of Composite Prepregs

Curing is an irreversible time-dependent and progressive molecular reaction process that requires heat energy, either by conventional heating or radiation, to raise molecular mobility [45]. Epoxy curing is an exothermic process that strengthens the material by cross-linking polymer chains [46]. Curing can be achieved by conventional heating, electron beams, chemical additives, or accelerated curing (e.g., microwave, radiofrequency, and ultra-violet radiation) [47]. The electromagnetic radiation cure mechanisms differ from thermal mechanisms in that curing is initiated by ionic or free radical intermediates formed by high-energy electromagnetic radiation [41].

The mechanism of microwave heating is shown in Figure 2. Epoxy resin is electrically neutral but dipolar in nature, as it possesses partial (+δ, −δ) charges. Microwave energy penetrated in a volumetric manner and increased the molecular mobility of resin due to periodic changing electrical fields (Figure 2). In this manner, the kinetic energy of molecules was increased and created a temperature gradient. Keeping in view the previous studies, the curing time and temperature were experimentally designed.

Five composite prepregs (of each proportion) were cured. “T″ refers to thermally cured, “M” to microwave cured, and “t_c_” to cure time. In the thermal cure method, the heating oven was raised to 40 °C then composite prepregs were put in (heating rate was 10 °C/min) to reach the curing temperature (160 °C). Curing times were 240 s, 300 s, 360 s, 420 s, 480 s, 600 s, and 660 s. In the microwave cure cycle, a set of composite prepregs was heated (300 W) for 20 s, 30 s 45 s, 60 s, 75 s, 90 s, and 100 s.

### 2.4. Characterization

The dielectric constant (Ɛr) and dielectric loss (δ) were measured (S-band) at PNA Network 8362B (Agilent) with a 3 mm inner bore circular disc (2 × 6 mm). The spectrums of thermally and microwave-cured silica fiber/epoxy composites (5 mg) were recorded at a resolution of 4 cm^−1^ using an FTIR spectrometer (Spectrum 100, Perkin-Elmer, Waltham, MA, USA). The microscopic morphology was obtained through scanning electron microscopy (SEM) (JSM-6490A, EOL, Tokyo, Japan) at an accelerating voltage of 20 kV. Calorimetric measurements were conducted on a DSC 6000, Perkin-Elmer, USA differential scanning calorimeter (DSC). 7 mg powder was placed in hermetic sample pan in the DSC cell which was raised from 50 °C to 250 °C (nitrogen atmosphere and heating rate of 10 °C/min). The storage modulus (E′), loss modulus (E″), and damping factor (Tan D) were obtained through a dynamic mechanical analyzer Q800 DMA (1 Hz and the heating rate at 5 °C/min. The weight loss was determined (30–800 °C) by thermogravimetric analysis on TGA Q600 SDT, (TA instruments, SHIMADZU, Kyoto, Japan at a heating rate of 20 °C under nitrogen atmosphere. The tensile and compression strength were measured (ASTM D3039) on the universal testing machine AGX-Plus (SHIMADZU, Kyoto, Japan) test speed of 2 mm/min using a 50 N load.

## 3. Results and Discussion

Dielectric performance, structural, morphology, thermal properties, and mechanical properties are discussed in this section.

### 3.1. Dielectric Properties

The reflection of electromagnetic (EM) waves on the surface and the energy loss inside the material is due to the loss of EM waves. Equations (1) and (2) determine the relative dielectric constant (Ɛ) and dielectric loss (δ) of dielectric material toward an electromagnetic field.
Ɛ_I_ = Ct/Ɛ_o_A(1)
δ = C/C_o_ω(2)
where Ɛ_I_ = dielectric constant, δ = dielectric loss C = capacitance with dielectric, t = sample thickness, Ɛo = permittivity of air (8.85 × 10^–12^ F/m), A = cross-sectional area of sample, C_o_ = capacitance without dielectric, and ω = angular frequency.

Dielectric permittivity describes how fast an electrical signal can transmit through a dielectric material and a low dielectric constant facilitates signal propagation across it. Dielectric constant and dielectric loss factors (with standard deviation, SD) of thermally cured and microwave-cured composites are mentioned in Table 1 and Table 2, respectively.

The Ɛr and δ of thermally cured composite were 3.90 and 0.053 (S_0.3_E_0.7_), 3.89 and 0.054 (S_0.4_E_0.6_), 3.80 and 0.050 (S_0.5_E_0.5_), 3.80 and 0.051 (S_0.6_E_0.4_), and 3.78 and 0.052 (S_0.7_E_0.3_). The dielectric properties (Ɛr and δ) decrease as the extent of the cure increases. From the results, both Ɛr and δ were decreased with the increase in cure time and decreased as the reaction progressed; the changes in the dielectric properties are related to the decreasing number of the dipolar groups in the reactants and the increasing viscosity [48].

The Ɛr and δ of microwave-cured composite were 3.81 and 0.046 (S_0.3_E_0.7_), 3.80 and 0.045 (S_0.4_E_0.6_), 3.77 and 0.043 (S_0.5_E_0.5_), 3.79 and 0.043 (S_0.6_E_0.4_) and 3.80 and 0.043 (S_0.7_E_0.3_). Since microwaves are high-energy waves, they increased the molecular mobility of epoxy molecules and raised the temperate. The rapid curing of epoxy resin was due to the epoxy–amine reaction progress to a greater extent than the epoxy–hydroxyl reaction [32]. Compared to the thermally cured composite, microwave-cured composite exhibited 1% lower Ɛr and 21.5% lower δ. It was due to the fast curing of thermosetting polymer with the progress of the epoxy–amine reaction.

Figure 3 shows the dielectric constant and loss of S_0.5_E_0.5,_ which was considered an optimum among all proportions. During curing, dielectric properties change due to the disappearance of epoxy, amine groups, and charge migration of dipolar groups [32]. Remarkable dielectric properties of silica fiber (x = 50%) composites were noted (i.e., dielectric loss of microwave cured). Adding silica fiber (x > 50%), the dielectric properties were insignificantly varied; however, decreasing x < 50%, the Ɛr, and δ were marginally increased. Considering the dielectric properties, S_0.5_E_0.5_, the structure, morphology, and thermal and mechanical properties were analyzed. Samples were ethanol washed, cleaned, air dried (2 h), and then kept in a desiccator at room temperature until required for testing.

### 3.2. Structure and Morphology

#### 3.2.1. FTIR Analysis

The FTIR spectral analysis of cured silica fiber/epoxy composites is shown in Figure 4. The spectrum shows characteristic absorption peaks of the epoxide ring between 400 cm^−1^ and 4000 cm^−1.^ The peak around 916 cm^−1^ assigned to the C–O deformation of the oxirane ring while the second band located at 1002 cm^−1^ represents the C–O–C stretching of the epoxy group and another band at 2922 cm^−1^, which is attributed to the C–H stretching of methylene group in oxirane. Reference peaks around 1631 cm^−1^ and 1504 cm^−1^ correspond to the C–C stretching vibration of aromatics, and the C=C stretching vibrations of -CH_3_, 2922 cm^−1^ are related to the C–H stretching vibration of CH_2_ and C–H stretching of –CH_3_, respectively [27,32]. A hydroxyl linkage is due to the -OH stretching band at 3426 cm^−1^. There is a decrease in the epoxy ring at 905 cm^−1^ and shows N–H compression at 1580 cm^−1^ confirming the reaction of epoxy resin through crosslinking of end epoxy groups with the hardener during curing [38]. The presence of an absorption peak at 1247 cm^−1^ represents the stretching of the C–N formed by cross-linking of the epoxy ring with an amine group hardener [18]. The peaks at 890 cm^−1^, 975 cm^−1^ and 1002 cm^−1^ are attributed to the presence of Si–OH compression, and Si–O–Si stretching vibrations. The spectra reveal the opening of the epoxide ring by an amine to form OH and CN groups and, conversion of epoxy groups. Comparable IR spectra of thermally and microwave-cured composites are found with similar functional groups, irrespective of the curing route.

#### 3.2.2. Morphology

SEM images in Figure 5 represent the surface morphology, fabric–matrix interaction, and fracture propagation, of S_0.5_E_0.5_-T600 (thermally cured). In the thermally cured composite Figure 5a–c, the resin is attached to fiber surfaces; however, fewer fibers are detached. In thermal curing, the heat is transferred through conduction from the outward surface to the inward, and in some portions, there might be different energy available for cure.

The SEM of the S_0.5_E_0.5_-M90 composite, as shown in Figure 5d–f, were similar but had a better fiber-matrix interaction than the thermally cured composite. During microwave curing, however, the irradiation and convection result in localized curing (cross-linking) of the thermoset resin. Despite the few voids there, the fibers were seen to be firmly intact with epoxy.

### 3.3. Thermal Properties

#### 3.3.1. Thermogravimetric Analysis (TGA)

Figure 6 shows the TGA curves where the red line shows a weight loss of thermally cured S_0.5_E_0.5_-T600 and the blue line represents microwave cured composite. The degradation was compared to 20% weight loss of the cured composite samples. In the thermally cured composite, degradation temperature was around 535 °C. An abundant weight loss (16.9%) was observed between 285 °C and 538 °C, which refers to pyrolysis and the maximum weight loss, was 27.1%. In the microwave-cured composite (S_0.5_E_0.5_-M90) there is a significant mass degradation that begins at 260 °C with a major weight loss (15.4%) from 268 °C to 548 °C. With the final wt. loss of 24.5%, the microwave-cured composite required more degradation energy than thermally cured composites. In a sense, the degradation temperature of microwave cured was slightly higher than that of the thermally cured composite.

Due to high energy radiation, the microwave energy efficiently cured composite than thermal heating. In thermal curing, mainly the heat was transferred to the composite surface, which was reached inside through conduction. Silica fibers are heat-resistance materials in nature that also affect conduction. However, microwave irradiations and convection end in quicker cross-linking of the epoxy in the composite. Due to this (2.6%) lower weight loss, microwave curing can be claimed as a superior route in manufacturing thermally stable composite.

#### 3.3.2. Differential Scanning Calorimeter (DSC)

DSC evaluated the thermal stability and phase transitions in silica fiber/epoxy composites, represented in Figure 7, where an increase in the glass transition temperature (Tg) was observed. Exothermic transitions appeared due to the polymerization of epoxy-amine. In a high-temperature region, more energy was available for the etherification of -OH and epoxy groups, destruction of weak bonds, and homo-polymerization of epoxide rings [26]. DSC curves of S_0.5_E_0.5_-T300, S_0.5_E_0.5_-T420, and S_0.5_E_0.5_-T600 are shown in Figure 7a, where transitions around 80 °C and near 200 °C indicate curing of thermosetting resin conversion of low mol. wt. monomers into a macromolecular cross-linked network through complex transformations [39]. With increased curing time, the height of exothermic transition peaks is more reduced than S_0.5_E_0.5_-T300 and S_0.5_E_0.5_-T420. The energy evolution is expected due to the possibility of side reactions and homo-polymerization of residual epoxide groups. S_0.5_E_0.5_-T600 (10 min cured) is found with the highest degree of increase in Tg, expected to have intermolecular interactions.

DSC of microwave-cured composite, Figure 7b, shows an increasing Tg, where the polymerization is initiated, propagated, and completed rapidly. The thermogram [S_0.5_E_0.5_ -M30 and S_0.5_E_0.5_ -M60] illustrates transitions around 204 °C to 208 °C, where the composite was cured in a short time as microwave irradiations boosted the reaction rate by more energy penetration and rapid increase in the system’s viscosity [33]. Microwave radiation has a more complex effect on the curing process than the temperature increase. Quick epoxy-amine crosslinking in a short time and microwave energy restricted the mobility of polymer chains [13]. From the thermogram of S_0.5_E_0.5_-M90 (cured for 90 s), it looks fully cured, with good dielectric properties, like that epoxy composites with low concentrations of primary amines [36].

### 3.4. Mechanical Properties

Viscoelastic behavior, ultimate tensile strength (UTS), and ultimate compression strength (UCS) of silica fiber/epoxy composite are discussed in this section.

#### 3.4.1. Dynamic Mechanical Analysis (DMA)

Viscoelastic properties of cured thermosetting polymers are often investigated using DMA as a standard technique suitable for application in a wide temperature range [48]. The thermomechanical spectra (cured composites) were obtained as shown in Figure 8a,b. Storage modulus (E′), loss moduli (E″), and mechanical damping (Tan **D**) as a function of temperature were presented. Storage moduli evaluate the material’s resistance to deformation while loss modulus quantifies the energy dissipation in the composites [49]. Tg of thermally cured composite (S_0.5_E_0.5_-T600) was 72.1 °C, storage modulus started dropping from 51.5 °C, and loss modulus from 64.0 °C.

Glass transition marks the thermal transition between glass and leathery regimes based on the peak of loss modulus [49]. The storage modulus also dropped before Tg; this referred to polymer chain mobility due to the increased thermal energy from the rising temperature. The chains in the epoxy matrix began to slide with higher degrees of freedom than below the first transition, usually referred to as beta transition. An increase in storage modulus, due to a change in T_β,_ refers to an increase in the energy dissipation ability of the composite.

DMA of microwave cured S_0.5_E_0.5_-M90, in Figure 8b shows 15.5% higher Tg (85.4 °C), and 20% modulus (storage and loss) than thermally cured composite. The EM waves in the microwave oven periodically changed the electric field, which raised the molecular mobility. This molecular motion increased the kinetic energy and the temperature of epoxy resin, where the quick progression of epoxy–amine polymerization caused a good, cross-linked composite. The DMA refers to the fact that the microwave cured offers better resistance against the applied stresses, which revealed superior stability than the thermally cured composite.

#### 3.4.2. Ultimate Tensile Strength (UTS)

Tensile stress–strain curves of thermally and microwave composites are shown in Figure 9. The UTS of S_0.5_E_0.5_-T600 was 75.79 MPa and that of S_0.5_E_0.5_-M90 was 89.68 MPa. UTS of the thermally cured was 15.4% lower than the microwave-cured composite. From the stress–strain curves in Figure 9, the tough nature of the composite is evident, as the microwave-cured specimen fractured at higher stress. This difference occurred due to the curing method. The strength of the epoxy becomes greater and reaches the maximum upon efficient crosslinking during curing [32]. The higher UTS indicates that the microwave composite structure became comparatively tough compared to the thermally cured composite due to the increased cross-linked composite density.

Composites fracture mechanism describes that epoxy resin transmitted the resistive forces on fibers and fractured as seen in Figure 10. SEM images of the thermally cured composite are shown in the Figure 10a and the microwave-cured composite is shown in Figure 10b. The applied load started cracks from the bonded epoxy resin on the surface, then the fiber–matrix interface, and finally the fibers. This scheme contributed to the final fracture of the composite structure.

The microwave-cured composite broke at a comparatively higher tensile load which can be seen in the Figure 10b that the fracture originated from the area where the fiber-to-matrix bonding was weak. However, a fractured specimen of the microwave-cured specimen showed an even load distribution, which resulted in higher tensile strength, the benefit of microwave curing over thermal heating.

#### 3.4.3. Ultimate Compression Strength (UCS)

In the compression test, the magnitude of opposing forces pushes inward on the specimen. The stress–strain curves showed a linear increase in the compression stress revealing a tough composite structure. The compression strength of S_0.5_E_0.5_-T600 (thermally cured) was 201.1 MPa and of S_0.5_E_0.5_-M90 (microwave cured) composite, was 210.25 MPa as shown in Figure 11. The compression strength of microwave composite was seen as 4.3% higher than that of the thermally cured composite. Initially, up to 5% strain, the thermally cured specimen offered more resistive forces, but then true stress–strain curves showed similar increasing trends for both samples. Microwave-cured specimens broke at a higher load than thermally cured composite.

Initially, S_0.5_E_0.5_-T600 seemed to be tougher than S_0.5_E_0.5_-M90 but as the curve progressed, its behavior was slightly changed. The applied force put a load on the surface and then transmitted it inward to the composite structure. In thermal curing, due to conduction, the composite outer surface was better cured than its inner structure. The strength of the epoxy became greater and reached maximum upon efficient crosslinking during curing [32]. Microwave-cured (S_0.5_E_0.5_-M90) was found with an improved compact structure than the thermally cured composite, as shown by the results and images.

SEM images of fractured specimens during the compression test, are shown in the Figure 12a,b. The low compression strength of thermally cured composite indicated that the fracture mechanism changed due to the lowering of the fiber–matrix interfacial shear in this specimen. The higher compression strength of S_0.5_E_0.5_-M90 revealed that microwave energy increased penetration and better cross-linking. It is clear from the SEM images that the thermally cured structure was more damaged than the microwave-cured composite. These results showed that microwave-assisted localized heating can improve the tensile and compressive strength of fiber-reinforced composites. However, change in curing conditions can improve mechanical properties with further experimentation.

## 4. Conclusions

The present work compared the effect of the curing route on the properties of silica fiber/epoxy composites. FTIR spectra revealed a similar chemical structure irrespective of the cure mechanism; however, microwave curing took a shorter time and less energy than thermal heating. A compact composite structure was seen where silica fibers were firmly embedded in epoxy resin. Composite cured through microwave energy was obtained with superior dielectric properties (1% lower dielectric constant, 21.5% lower dielectric loss factor), higher thermal stability (2.6% lower wt. loss), and better mechanical properties (higher Tg, storage, loss modulus, 15.4% higher tensile strength, and 4.3% higher compression strength) than thermally cured composite. In microwave heating, high-energy electromagnetic radiations increase the molecular mobility of dipolar epoxy molecules and facilitate efficient curing. The increased kinetic energy of molecules provides a high temperature to put a complex effect on curing compared to conduction in thermal curing. This study suggests that microwave curing is a superior alternative to thermal curing for achieving low dielectric constant, dielectric loss factor, fair thermal stability, and mechanical properties of silica fiber/epoxy composites for microelectronics. Further research is recommended to optimize microwave curing conditions to attain even better results.

## Figures and Tables

**Figure 1 materials-16-01790-f001:**
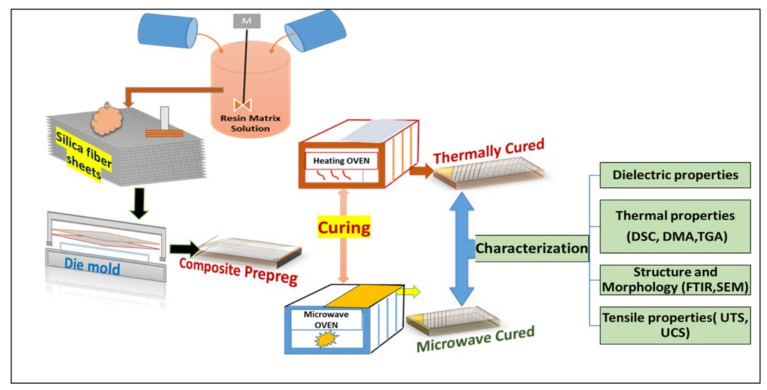
Schematic (raw material to prepreg to composite).

**Figure 2 materials-16-01790-f002:**
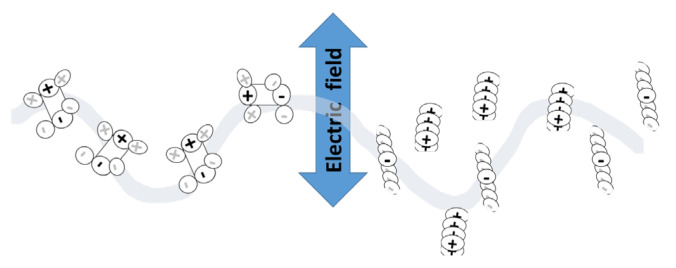
Microwave heating mechanism.

**Figure 3 materials-16-01790-f003:**
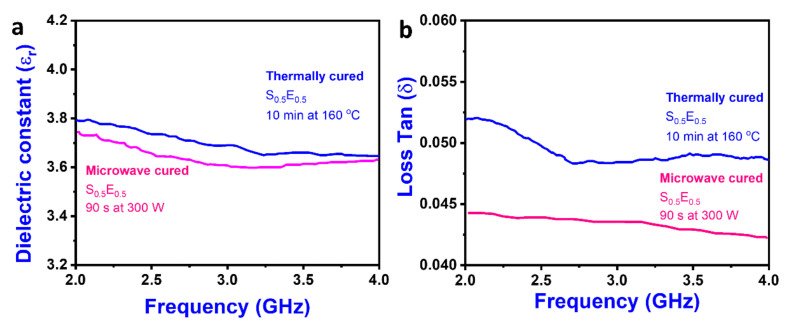
(**a**) Dielectric constant and (**b**) loss factor of the cured composites.

**Figure 4 materials-16-01790-f004:**
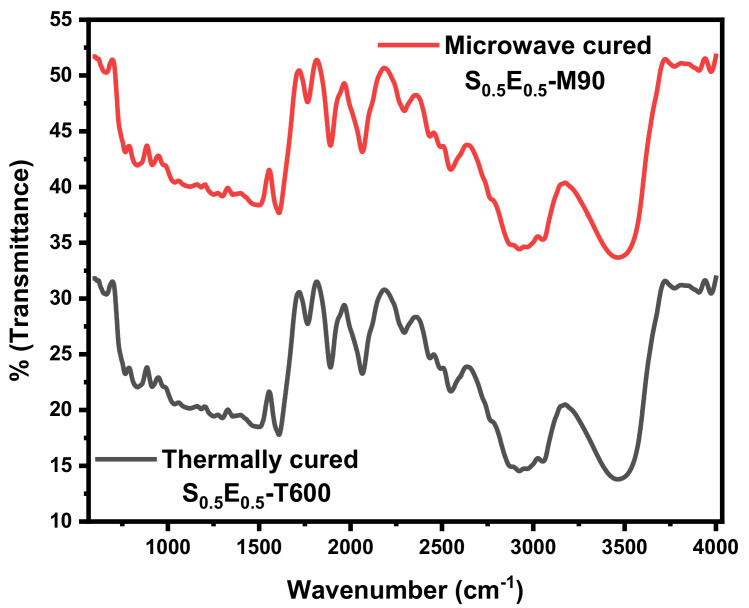
FTIR of the cured composite.

**Figure 5 materials-16-01790-f005:**
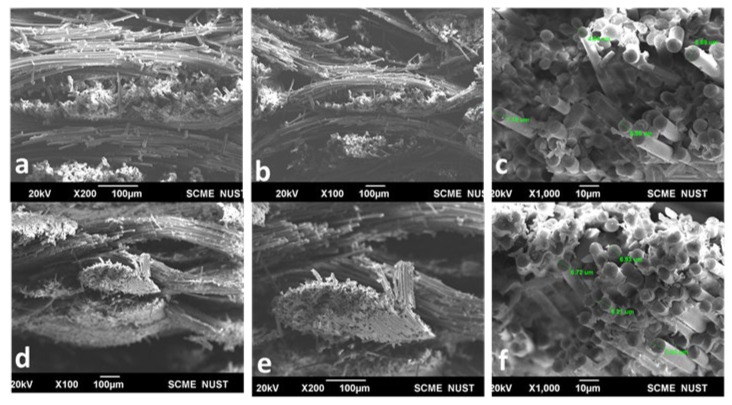
Morphology of (**a**–**c**) thermally and microwave (**d**–**f**) cured composites.

**Figure 6 materials-16-01790-f006:**
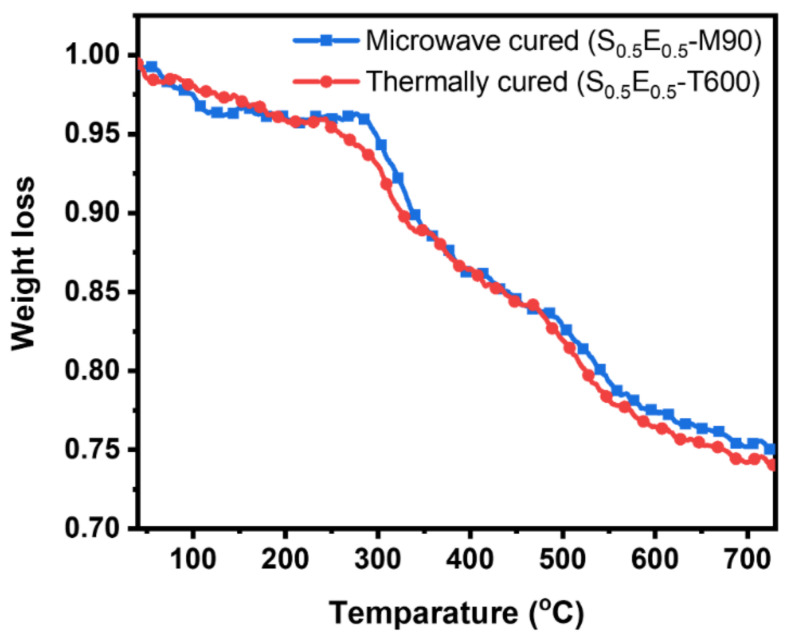
TGA of thermally and microwave-cured composite.

**Figure 7 materials-16-01790-f007:**
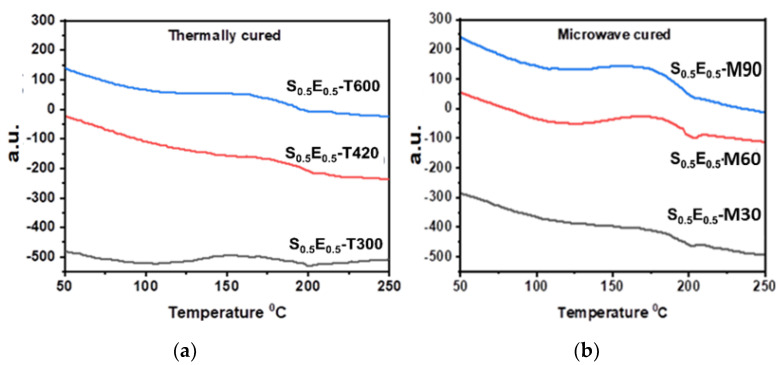
DSC of (**a**) thermally and (**b**) microwave-cured composite.

**Figure 8 materials-16-01790-f008:**
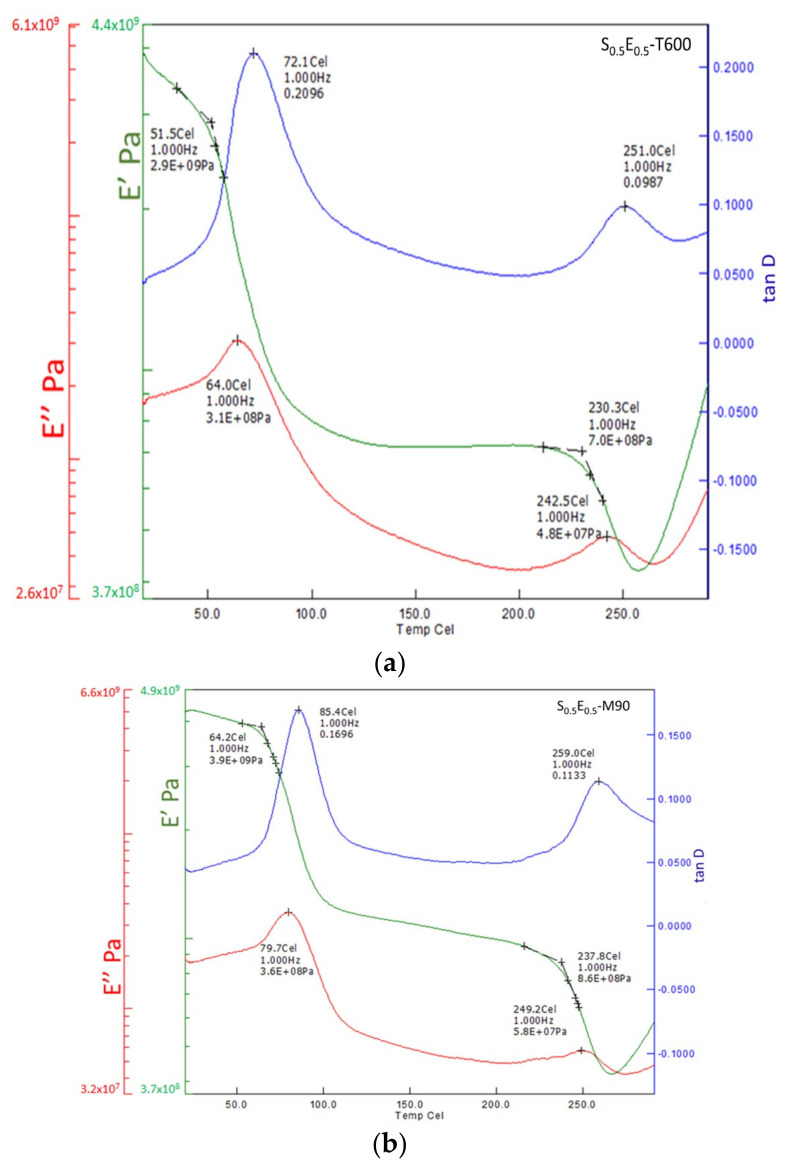
DMA plot (**a**): thermally cured composite. (**b**): microwave cured composite.

**Figure 9 materials-16-01790-f009:**
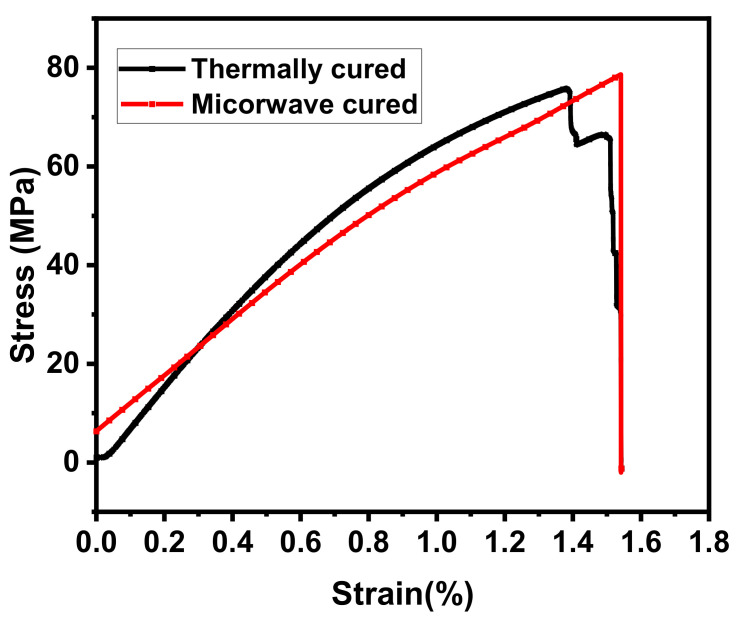
Tensile strength of cured composites.

**Figure 10 materials-16-01790-f010:**
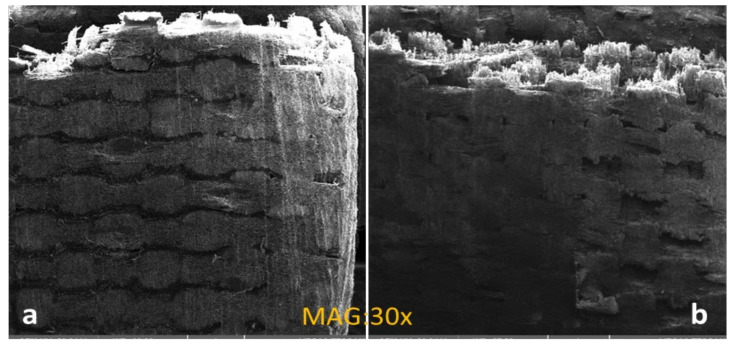
Fractured in tensile test (**a**) thermally cured (**b**) microwave cured.

**Figure 11 materials-16-01790-f011:**
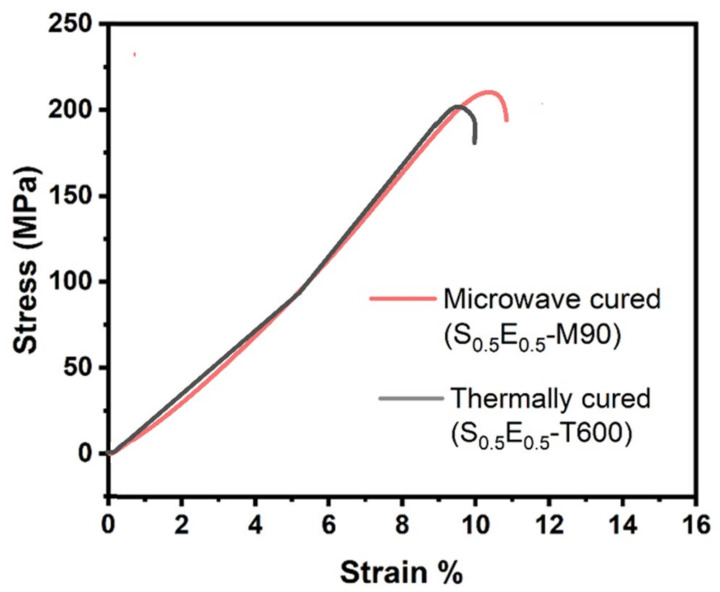
Compression strength of cured composites.

**Figure 12 materials-16-01790-f012:**
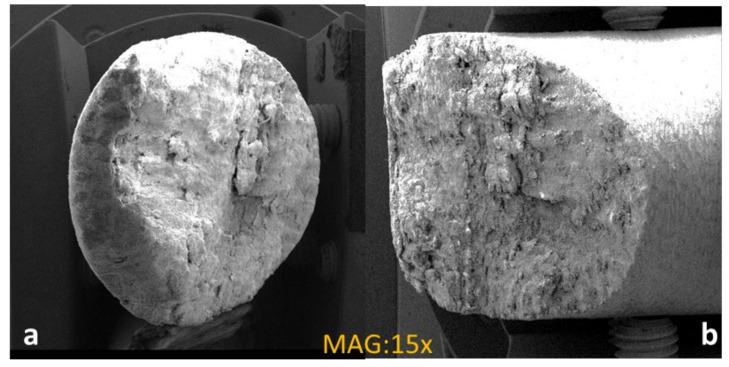
Compression fractured (**a**) thermally cured (**b**) microwave cured.

**Table 1 materials-16-01790-t001:** Dielectric properties of thermally cured composite at 160 °C.

t_c_	S_0.3_E_0.7_		S_0.4_E_0.6_		S_0.5_E_0.5_		S_0.6_E_0.4_		S_0.7_E_0.3_	
(sec)	Dk	Loss	Dk	Loss	Dk	LOSS	Dk	Loss	Dk	Loss
240	3.98	0.059	3.98	0.057	3.96	0.057	3.96	0.057	3.94	0.057
SD	0.006	0.001	0.010	0.002	0.0292	0.001	0.029	0.001	0.0248	0.002
300	3.96	0.056	3.97	0.056	3.95	0.055	3.95	0.054	3.91	0.056
SD	0.010	0.001	0.010	0.001	0.010	0.001	0.031	0.001	0.002	0.001
360	3.94	0.056	3.95	0.053	3.85	0.053	3.89	0.053	3.89	0.054
SD	0.010	0.001	0.010	0.002	0.048	0.002	0.039	0.001	0.027	0.002
420	3.91	0.053	3.91	0.055	3.83	0.053	3.85	0.053	3.83	0.053
SD	0.008	0.001	0.008	0.002	0.001	0.051	0.001	0.001	0.049	0.001
480	3.91	0.052	3.89	0.052	3.81	0.051	3.82	0.052	3.79	0.052
SD	0.014	0.001	0.019	0.001	0.001	0.001	0.010	0.001	0.025	0.002
600	3.90	0.053	3.89	0.054	3.80	0.050	3.82	0.051	3.78	0.052
SD	0.014	0.001	0.015	0.001	0.061	0.001	0.001	0.001	0.014	0.001
660	3.90	0.053	3.98	0.054	3.79	0.051	3.80	0.051	3.94	0.052
SD	0.018	0.001	0.002	0.012	0.002	0.008	0.002	0.001	0.010	0.001

**Table 2 materials-16-01790-t002:** Dielectric properties of microwave-cured composite at 300 W.

t_c_	S_0.3_E_0.7_		S_0.4_E_0.6_		S_0.5_E_0.5_		S_0.6_E_0.4_		S_0.7_E_0.3_	
(sec)	Dk	Loss	Dk	Loss	Dk	Loss	Dk	loss	Dk	Loss
20	3.98	0.050	3.96	0.051	3.92	0.048	3.94	0.048	3.94	0.048
SD	0.023	0.001	0.012	0.005	0.005	0.002	0.008	0.002	0.009	0.003
30	3.89	0.049	3.89	0.049	3.89	0.046	3.90	0.047	3.93	0.046
SD	0.042	0.001	0.027	0.003	0.020	0.003	0.020	0.001	0.024	0.002
45	3.86	0.048	3.85	0.048	3.85	0.045	3.86	0.045	3.88	0.046
SD	0.036	0.001	0.013	0.001	0.043	0.004	0.029	0.002	0.034	0.002
60	3.84	0.048	3.82	0.046	3.82	0.044	3.82	0.043	3.84	0.043
SD	0.026	0.002	0.026	0.001	0.026	0.00	0.009	0.002	0.034	0.002
75	3.80	0.046	3.79	0.045	3.78	0.043	3.79	0.042	3.80	0.043
SD	0.017	0.001	0.030	0.002	0.053	0.004	0.036	0.002	0.045	0.001
90	3.81	0.046	3.80	0.045	3.77	0.043	3.79	0.043	3.80	0.043
SD	0.041	0.003	0.026	0.001	0.060	0.002	0.023	0.002	0.028	0.003
100	3.80	0.046	3.80	0.051	3.78	0.043	3.79	0.041	3.80	0.042
SD	0.045	0.004	0.038	0.002	0.033	0.003	0.027	0.002	0.046	0.002

## Data Availability

Not applicable.

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
