# Peer review of "Silica-Fiber-Reinforced Composites for Microelectronic Applications: Effects of Curing Routes"

_materials, 2023, doi:10.3390/ma16051790_

Round 1

Reviewer 1 Report

The authors compared thermal and microwave curing on silica fiber composites. The interest of microwave has been demonstrated previously and deserves to be highlighted. According to the results, microwave curing has better mechanical properties than thermally produced materials which  is attributed to the higher cross-linked density. Though the findings are consistent with the results, serious changes are needed, both in form and substance, in order to be published. Please take into consideration the following remarks:

  1. The microwave experiment are not clearly described and it raises questions. The use of microwave oven is an interesting topics but the reproductibilty must be investigated. In this sense, standard deviations are needed (tables 3, 4 and figure 2).
  2. The curing time for both methods are not clearly expressed. This is necessary to support comment (lines 221 to 226) and conclusion.
  3. Specify the FTIR resolution.
  4. Correct line 199 (table 3 not table 1).
  5. Line 208: the ref is not adapted.
  6. In the thermal section, the ATG experiments should be presented before the DSC and the DMA analysis moved to a mechanical section.
  7. The Ea should be extracted from the DSC experiments and discussed to support the chemical reactivity comment.
  8. The section on structure and morphology should be placed before the calorimetric results.
  9. Correct line 389 (fig. d-f)
  10. Correct figure 8.

Author Response

Q.1. The microwave experiment are not clearly described, and it raises questions. The use of microwave oven is an interesting topic, but the reproducibility must be investigated. In this sense, standard deviations are needed (tables 3, 4 and figure 2).

Response:

Thank you for reviewing our manuscript, and highlighting the discrepancies for improvement.

Microwave experiments were organized based on previous studies (line: 130-145, line 146, and line 151 section 2.3)  

Five composite prepregs (of each proportion) were cured (line: 46 section 2.3).

Due to the extensive experimental data, table 1 and table 2 are being modified now (lines 185 and 189 section 3.1). Standard deviations are added (table 1 and table 2 section 3.1).

Q.2 The curing time for both methods are not clearly expressed. This is necessary to support comment (lines 221 to 226) and conclusion.

Response:

Keeping the previous studies and authors' experience, the curing conditions were designed in this study (line; 130-145, line 146 and line 148-151, section 2.3, line 242 section 3.2.2) and expressed as:

In thermal heating, curing of composite prepregs was mainly due to conduction. Thermal energy flows or transfers from the outer surface to the inward.

Microwave Heating

Electromagnetic radiation cure mechanisms differ from thermal mechanisms in that curing is initiated due to high energy electromagnetic radiation [21, 40]. Epoxy resin is electrically neutral but dipolar in nature, it possesses partial +, - charges. The periodically changing electrical field due to microwave energy increases the molecular mobility of the resin. Microwave energy penetrates in a volumetric manner, which increases the kinetic energy, creating a temperature gradient for the curing of thermoset resin. It is now defined clearly (line: 66- 76, section 1, and line: 135-144 and fig.2 section 2.3).

To support the comment (lines 221 to 226)

  1. Lines 221-226 now line 196-200,
  2. Composite dielectric properties are based on constituent materials. In this study, the fabricated composite has a low dielectric constant (Ɛr) and dielectric loss (δ). We attempted to select/fabricate the proportion silica fiber/epoxy composite with optimum dielectric performance, structural, morphology, thermal properties, and mechanical properties) were corrected and edited in the manuscript (line197- 199, 208-212 section 1).

Q.3. Specify the FTIR resolution.

Response:

The FTIR resolution was 4 cm-1 (mentioned now in line 156 section 2.4)

Q.4. Correct line 199 (table 3 not table 1).

Response:

Thanks for highlighting the mistake.

The table number was corrected (mentioned now in lines 184 and 188 section 3.1)

Q.5. Line 208: the ref is not adapted.

Response:

Agreed and thank you for pointing it out.

The correction was made (mentioned now in lines 194-198 section 3.1).

Q.6. In the thermal section, the TGA experiments should be presented before the DSC & DMA analysis moved to a mechanical section.

Response:

In the thermal properties, the sequence was changed as per the reviewer's suggestion

TGA results are now presented before DSC (line 253, section 3.3.1).

DMA results are now written in the mechanical properties section (line 306, section 3.4.1).

Q.7. The Ea should be extracted from the DSC experiments and discussed to support the chemical reactivity comment.

Response:

For prepregs, semi-cured, and un-cured epoxy composites, the DSC graph provides information from the energy transitions due to heat supplied in the test. In this study, DSC (50-250 oC), there was insignificant information due to limited transitions in cured composites. Silica fiber/epoxy composites were maximum or completely cured in both curing mechanisms as represented by energy changes, However, the smaller energy shift in thermally cured (T 420 and T 480) and microwave cured (M 210 oC to M 208 oC) were there.

TGA analysis showed a higher wt. loss (%) in thermally cured than a microwave (MW) cured composite. The thermomechanical response of MW cured against the stresses was found better than the thermally cured composite. Keeping in view of TGA, DSC, and DMA, thermal properties were discussed (line 252, sections 3.3 and 3.4.1).

Q.8. The section on structure and morphology should be placed before the calorimetric results.

Response:

The authors agree with the reviewer’s comment.

Structure and morphology are now mentioned before calorimetric results.  

FTIR (line 218, section 3.2.1).

Morphology (line 238, section 3.2.2).

Q.9. Correct line 389 (fig. d-f)

Response:

Thank you for highlighting this shortcoming.

The fig.(d-f) was corrected (line 256 section 3.2.2)

Q.10.      Correct figure 8

Response:

Figure 8 has now been corrected.

Previous Fig. 8 (Stress-strain curve of the tensile test) is corrected and now it is Fig. 9 (line 349 section 3.4.2) 

Also, previous Fig.10 (stress-strain curve of compression test) is corrected and is now fig10 (line 377 section 3.4.3).

Reviewer 2 Report

Micro wave curing seems to be slightly better than the conventional curing. But the reason is not clear. It has to be discussed based on cure kinetics.

Also it is better to bring out the other advantages of Micro wave curing, especially the cost involved.

Author Response

Q.1. Microwave curing seems to be slightly better than the conventional curing. But the reason is not clear. It must be discussed based on cure kinetics.

Response:

Thank you for reviewing our manuscript and highlighting the discrepancies and improvement points.

In conventional curing, composite prepregs were heated through conduction. The heat flow or heat transfer is from the outer surfaces to the inward structure (composite prepreg). 

The electromagnetic radiation cure mechanisms differ from thermal mechanisms in that curing is initiated due to high energy electromagnetic radiation [21, 40]. Epoxy resin possesses dipole movement. Their molecular mobility increases, due to in-depth penetration of microwave energy, which increase the kinetic energy. Thus, high temperature /energy is available for efficient curing. It is now defined clearly (line: 66- 76, section 1, and line: 135-144 and fig.2 section 2.3).

Q.2. Also, it is better to bring out the other advantages of Microwave curing, especially the cost involved.

Response:

Thank you for highlighting another important point The reviewer’s comment will bring more quality to the publication.  Advantages of microwave curing are:

  1. Reduce production cost (line 52 section 1)
  2. Direct heating of samples leads to efficient, fast, and selective heating compared  to  conventional  thermal  curing  where  the  samples  are  heated  indirectly  (line 55-56 section 1)
  3. Microwave radiation heats in a volumetric manner, leading to less temperature disparity and therefore a less severe temperature gradient within the material, leading potentially to less internal residual stress (lines 57-58 section 1).
  4. Reducing 45% processing time and 3% energy consumption (line 61 section 1)
  5. Much faster curing (shorter gelation and vitrification times) and higher final Tg of the resulting materials (particularly for DGEBA). line 66-77 section 1
  6. Comparable or better electrical, thermal, and mechanical properties (section 3)

Advantages of Microwave curing (cost involved).

We have added the introduction, results, and discussion sections. (Sections 1 and 3).

In this study, less time and energy are consumed in microwave curing than in thermal curing. Complete cost estimation requires the precise calculation of energy consumption, losses, and waste. A controllable accelerated microwave needs to be developed via controlling temperature gradient, to see the significant economic impact on composites manufacturing. In conclusion, it is mentioned as a suggestion for the future.

Reviewer 3 Report

The paper appears of interest, however there are a few flows that need to be addressed.

The main issue is related with the experiments, which results must be related to a larger experimental campaign. At least three specimens shall be tested every time, to get a minimum of statistical significance.

Furthermore English is required to undergo a deep review.

Further comments are reported in the attached.

Author Response

Q.1. The main issue is related with the experiments, which results must be related to a larger experimental campaign. At least three specimens shall be tested every time, to get a minimum of statistical significance.

Response:

Thank you for reviewing our manuscript, with your valuable feedback.

  1. Microwave experiments were organized based on previous studies (line: 130-145, line 146, and line 151 section 2.3)
  2. Five composite prepregs (of each proportion) were cured (line: 46 section 2.3).
  • Due to the extensive experimental data, table 1 and table 2 are being modified now (lines 185 and 189 section 3.1). Standard deviations are added (table 1 and table 2 section 3.1).
  1. Composite dielectric properties are based on constituent materials. In this study, the fabricated composite has a low dielectric constant (Ɛr) and dielectric loss (δ). We attempted to select/fabricate the proportion silica fiber/epoxy composite with optimum dielectric performance, structural, morphology, thermal properties, and mechanical properties (line197- 199, 208-212 section 3.1).

Q.2. English is required to undergo a deep review.

.Response:

Thank you again for the detailed, comprehensive, and critical manuscript review by highlighting deficiencies. This will bring quality improvements to our manuscript. The manuscript was again reviewed by all authors in light of improving the quality of the language used.

Round 2

Reviewer 2 Report

accept